# Identification of Virulence Factors in Isolates of *Candida haemulonii*, *Candida albicans* and *Clavispora lusitaniae* with Low Susceptibility and Resistance to Fluconazole and Amphotericin B

**DOI:** 10.3390/microorganisms12010212

**Published:** 2024-01-20

**Authors:** Letizia Angiolella, Florencia Rojas, Andrea Giammarino, Nicolò Bellucci, Gustavo Giusiano

**Affiliations:** 1Department of Public Health and Infectious Diseases “Sapienza”, University of Rome, Piazzale Aldo Moro 5, 00185 Rome, Italy; andrea.giammarino98@gmail.com (A.G.); bellucci.nico99@gmail.com (N.B.); 2Departamento de Micología, Instituto de Medicina Regional, Universidad Nacional del Nordeste, CONICET, Resistencia 3500, Argentina; florenciarojjas@hotmail.com (F.R.); gustavogiusiano@yahoo.com.ar (G.G.)

**Keywords:** *C. albicans*, *C. lusitaniae*, *C. haemulonii*, resistance, virulence factors, biofilm, adherence, hydrophobicity, stress, *Galleria mellonella*

## Abstract

Emerging life-threatening multidrug-resistant (MDR) species such as the *C. haemulonii* species complex, *Clavispora lusitaniae* (sin. *C. lusitaniae*), and other *Candida* species are considered as an increasing risk for human health in the near future. (1) Background: Many studies have emphasized that the increase in drug resistance can be associated with several virulence factors in *Candida* and its knowledge is also essential in developing new antifungal strategies. (2) Methods: Hydrophobicity, adherence, biofilm formation, lipase activity, resistance to osmotic stress, and virulence ‘in vivo’ on *G. mellonella* larvae were studied in isolates of *C. haemulonii*, *C. albicans*, and *C. lusitaniae* with low susceptibility and resistance to fluconazole and amphotericin B. (3) Results: Intra- and interspecies variability were observed. *C. haemulonii* showed high hydrophobicity and the ability to adhere to and form biofilm. *C. lusitaniae* was less hydrophobic, was biofilm-formation-strain-dependent, and did not show lipase activity. Larvae inoculated with *C. albicans* isolates displayed significantly higher mortality rates than those infected with *C. haemulonii* and *C. lusitaniae*. (4) Conclusions: The ability to adhere to and form biofilms associated with their hydrophobic capacity, to adapt to stress, and to infect within an in vivo model, observed in these non-wild-type *Candida* and *Clavispora* isolates, shows their marked virulence features. Since factors that define virulence are related to the development of the resistance of these fungi to the few antifungals available for clinical use, differences in the physiology of these cells must be considered to develop new antifungal therapies.

## 1. Introduction

Due to the increase in the number of multidrug-resistant (MDR) microorganisms and the unavailability of new antimicrobials, multidrug resistance (MDR) is recognized as one of the most important public health threats of the 21st century [1]. The World Health Organization (WHO) warns that MDR will constitute one of the leading causes of death in the world, causing approximately 10 million deaths annually in 2050 [2]. Fungi represent a growing clinical threat due to the limited availability of drugs [3]. Acquired antifungal resistance is related to stress response activation such as adaptive mechanisms, drug target modification, or an overexpression of or increase in multidrug transporters [4]. Emerging life-threatening multidrug-resistant *Candida* species such as *C. auris*, the *C. haemulonii* species complex, *C. glabrata*, *Clavispora lusitaniae* (sin. *C. lusitaniae*), *Meyerozyma guilliermondii* (sin. *C. guillermondii*), and the *C. parapsilosis* species complex are considered as an increasing risk for human health in the near future [5,6].

*C. haemulonii* complex candidiasis is associated with superficial to invasive infections in patients with risk factors like peripheral vascular disease, diabetes mellitus, organ transplants, malignant tumors, and chronic leg ulcers [7]. Knowledge about its biology and virulence factors is scarce. Its adaptive ability which contributes to survival in different host niches was observed. In addition, its capability to evade the action of commonly used antifungal agents, especially azoles and amphotericin B, makes this complex a worrying emerging opportunistic pathogen for either immunocompromised or immunosuppressed humans [8,9,10].

*C. lusitaniae* was first documented in 1979 and has attracted attention because it exhibits the development of resistance to amphotericin B, 5-fluorocytosine, or fluconazole [11,12,13]. *C. lusitaniae* was reported as being responsible for approximately 19.3% of fungemia in patients with cancer and 1.7% of genitourinary candidiasis cases in ambulatory patients, and it has also been associated with peritonitis and meningitis [13,14,15].

*Candida* drug resistance can be associated with several virulence factors [16]. In *C. albicans*, the main virulence factors are cell wall barriers, adherence, dimorphism, biofilm formation, proteins related to stress tolerance, hydrolytic enzymes (such as proteases, lipases, and hemolysins), and toxin production [17,18]. Adherence contributes to the persistence of the organism within the host, and this virulence factor is thus considered to be essential for the fungus settling and spreading [19]. Regardless of the species, microbial adherence due to hydrophobic interaction depends on the microbe surface hydrophobicity [20]. Likewise, biofilm formation is an important virulence feature that favors fungal pathogenesis. *Candida* species form a multicellular biofilm which allows yeast cells to anchor to host tissues, catheters, implants, and other devices [21]. It has been highlighted that increasing drug resistance has provided a strong impetus to understand the mechanisms of the increased tolerance of biofilm-associated infections to aid antimicrobial therapy.

Among the main virulence mechanisms of *Candida*, the ability to secrete hydrolytic enzymes like phospholipase C, caseinase, lipase, and proteinase stands out, to degrade the host cellular components, in order to facilitate penetration and invasion, thus playing a determining role in the pathogenesis [22]. These virulence factors have not been determined for all species and are not well known. Understanding and assessing these virulence factors in emerging drug-resistant yeast pathogens will contribute to the future development of novel target-specific drugs. In this study, we have evaluated hydrophobicity, the ability to adhere to and form biofilm, lipase activity, resistance to osmotic stress, and infection in an in vivo model of isolates of *C. albicans*, *C. haemulonii*, and *C. lusitaniae* with low susceptibility and resistance to fluconazole and amphotericin B.

## 2. Materials and Methods

### 2.1. Microorganisms and Growth Conditions

Three vaginal isolates of *C. albicans* (IMR-M-L 1462, IMR-M-L 1463, and IMR-M-L 1464), three blood isolates of *C. haemulonii* (IMR-M-L 785, IMR-M-L 1293, and IMR-M-L 1375), and four *C. lusitaniae* (IMR-M-L 301 and IMR-M-L 1112 isolated from blood and IMR-M-L 522 and IMR-M-L 1384 isolated from urine) were studied. *Candida albicans* ATCC 24433 was included as the quality control strain. All isolates were identified using the Vitek matrix-assisted laser desorption ionization–time of flight mass spectrometry (MALDI-TOF MS) system (Bruker Daltoniks, Bremen, Germany) and deposited in the culture collection (IMR-M-L) of the Mycology Department, Instituto de Medicina Regional, Universidad Nacional del Nordeste. Prior to studying the virulence factors, all yeast isolates were grown on Sabouraud Dextrose (SD) agar.

### 2.2. Antifungal Activity of Fluconazole and Amphotericin B

The antimicrobial activity of fluconazole and amphotericin B against all clinical and standard strains were evaluated using the broth microdilution method proposed by the reference CLSI M27-A4 methodology [23] and interpretative breakpoints were used according to the CLSI M27M44S [24].

Amphotericin B and fluconazole clinical breakpoints were not established for *C. haemulonii* and *C. lusitaniae*. However, epidemiological cut-off points (ECVs) were established for fluconazole against *C. haemulonii* and *C. lusitaniae* and also for amphotericin B against *C. albicans* and *C. lusitaniae* [25].

### 2.3. Hydrophobicity Assay

The cellular surface hydrophobicity (CSH) was determined by a two-phase system following previous reports [26]. In brief, yeasts were grown in SD broth at 28 °C for 24 h. Then, cells were washed with sterile saline buffer and 0.5% Tween 20 was added; they were resuspended in 0.05 M PBS (pH 7.2) at a final concentration of 2 × 10^6^ cells/mL. The cell suspension was transferred to a glass tube containing 500 μL octane (Sigma Aldrich, Saint Louis, MO, USA). The mixture was vortexed for 1 min and maintained at room temperature for phase separation. After the two phases had been separated, the aqueous phase was measured at OD600. The group without the octane overlay was used as the control. Relative CSH was calculated as follows: [(OD600 of the control-OD600 after octane overlay)/OD600 of the control] × 100. The value for each strain was the average of three independent biological replicates.

### 2.4. Adherence on Plastic Surface by Crystal Violet Assay

The ability of *Candida* isolates to adhere on polystyrene surface were measured as reported previously [27]. The yeast cells were grown as described above for 24 h at 28 °C, washed twice with sterile PBS, and then resuspended at 37 °C in RPMI 1640 plus 10% FBS at 2.5 × 10^7^ cells. After incubation for 3 h at 37 °C in six-well polystyrene plates (Corning Incorporated, Corning, NY, USA), the medium was aspirated and the non-adherent cells were removed and washed with PBS. The adherent cells were fixed with 99% *v*/*v* methanol for 15 min and then 0.02% (*v*/*v*) crystal violet was added for 20 min. Cells were washed and 33% (*v*/*v*) acetic acid was added again for 30 min. The crystal violet released was measured at 590 nm.

### 2.5. Biofilm Formation

Following previously described studies, the in vitro biofilm formation assay was carried out [28]. A yeast cell suspension (1.0 × 10^7^ cells/mL) was incubated for 24 and 48 h at 37 °C in 96-well microtiter plates (Corning, NY, USA). After formation of the mature biofilm, the medium was aspirated, and non-adherent cells were removed by washing with sterile PBS. The biofilm development was measured with a 2,3-bis-(2-methoxy- 4-nitro-5- sulfophenyl)-2H-tetrazolium-5-carboxanilide (XTT) reduction assay, a reaction catalyzed by mitochondrial dehydrogenases. In brief, biofilm cells were washed with PBS and then incubated with 0.5 mg/mL of XTT and 1 μM of menadione in PBS at 37 °C for 2 h. A sample (500 µL) was then transferred from each well into a fresh 12-well plate and the colorimetric change resulting from XTT reduction was measured at 490 nm.

### 2.6. Lipase Activity Assay

The lipase activity (Lz) was assessed according to Muhsin et al. [29]. Briefly, 10 µL of each strain suspension was cultured in a sterile Petri dish containing a lipid medium (i.e., peptone 1%, sodium chloride 5%, calcium chloride 0.01%, and agar 2%, plus 1% tween 80) and incubated at 32 °C for 20 days. A clear halo zone of precipitation around the colony indicated lipase production. The production of lipase was expressed as a ratio of diameter of a colony to total diameter plus zone of precipitation. Each test was performed in duplicate and the results were expressed as the average of the two obtained values. The ranges of activity according to the Lz index were established as follows: high, Lz ≤ 0.69; moderate, Lz = 0.70–0.89; weak, Lz = 0.90–0.99; none, Lz = 1.

### 2.7. Sensitivity to Osmotic Stress in Sodium Chloride

With some modifications to determine the sensitivity of *Candida* isolates to NaCl, the method reported by Chaves and da Silva [30] was applied. Ten microliter volumes of Sabouraud-grown yeast cells were transferred to 100 µL of SD broth, with the addition of 0.03–30% NaCl in 96-well microtiter plates, and incubated at 32 °C for 48 h. Growth was determined when visually perceptible turbidity was observed within each well.

### 2.8. Infection in Galleria Mellonella Model

For survival analyses, 10 larvae (250–320 mg/each) of *G. mellonella* per each *Candida*/*Clavispora* isolate for testing and two control groups were employed. Larvae were previously incubated at 37 °C. Larvae were selected at random for each group in the procedure and injected with different concentrations of yeast cells (about 5 × 10^7^ cells CFU/mL) into the hemocoel through the last left proleg (Hamilton syringe 701N; volume, 10 μL; needle size, 26 s; cone tip; Sigma-Aldrich, Milan, Italy) [31]. After injection, larvae were incubated in Petri dishes at 37 °C in standard aerobic conditions and survival was recorded at 24 h intervals for five days. Larvae were considered dead when they showed no movement in response to gentle prodding with a pipette tip. As a control, one group that did not receive injection and one group that was injected with phosphate-buffered saline (PBS) plus 0.01% Tween 20 were included. Each experiment was repeated three times.

### 2.9. Statistical Analysis

Statistical differences among the groups of data were analyzed by two-way ANOVA. In all of the comparisons, a *p* value of 0.05 or lower was considered to be significant. The analyses were performed using GraphPad Prism Software version 8.0.1.

## 3. Results

### 3.1. Susceptibility to Fluconazole and Amphotericin B

Table 1 shows the antifungal activity of fluconazole against the eleven isolates included in this work. The MIC values ranged from 0.25 (μg/mL), obtained for the reference strain, to >64 (μg/mL).

The values obtained for QC *Candida albicans* ATCC 24433 were within the ranges established by the reference document. All clinical isolates of C. albicans presented decreased susceptibility to fluconazole (*C. albicans* IMR-M-L 1462 was resistant and the other was susceptible-dose-dependent).

### 3.2. Hydrophobicity Assay and Adherence on Plastic Surface

The CSH and the ability to adhere to plastic surfaces are two important virulence factors to initiate infection and biofilm formation. Figure 1A shows the histograms, in percentage, of the CSH obtained for all of the strains of the three species studied. Variability in CSH was observed between the different species. In fact, the statistical analysis of all strains of each species shows significant differences **** (*p* < 0.0001) between *C. albicans* and *C. lusitaniae*, while differences in CSH were not significant for *C. haemulonii* (Appendix A). *C. lusitaniae* was the species with the lowest hydrophobicity in contrast to the other species studied.

The ability of *C. albicans*, *C. lusitanieae*, and *C. haemulonii* isolates to adhere to the polystyrene surface is shown in Figure 1B. All strains were highly adherent to the polystyrene surface. Differences between the three species were observed. The statistical analyses of all strains of each species show that *C. albicans* is the more adherent species with respect to *C. haemulonii* *** (*p* < 0.0003) (Appendix A).

### 3.3. Biofilm Formation

Histograms of biofilm formation after 24 and 48 h measured using XTT for Candida and Clavispora isolates (IMR-M-L) and the reference strain ATCC 24433 *C. albicans* are shown in Figure 2. All clinical isolates of the three species were able to produce biofilm with intraspecies variations, as depicted in Figure 2.

The statistical analysis of all strains of each species indicated significant dissimilarities between *C. albicans* and *C. haemulonii* during the initial 24 h of biofilm formation, where *C. haemulonii* was the lower biofilm producer with *** *p*-values < 0.0004 (Appendix A). Nonetheless, these variations decreased after 48 h of incubation. A noticeable difference was observed between *C. albicans* and *C. lusitaniae*, with a * *p*-value < 0.0392 (Appendix A).

### 3.4. Lipase Activity and Osmotic Stress

Table 2 reports the lipase activity expressed as the LZ index in *Candida* spp. Positive lipase activity (Lz < 1) was measured in 63% of the isolates studied. Lz values ranged from 0.32 to 0.70. Strong lipase production (Lz ≤ 0.69) was observed for five isolates (45.45%), *C. albicans* (IMR-M-L 1462, 1463, 1464) and *C. haemulonii* (IMR-M-L 1293 and 1345) and in the reference strain *C.albicans* ATCC 24433. Only *C. lusitaniae* IMR-M-L 1384 showed moderate activity (Lz = 0.70 ± 0.00).

No enzymatic activity (Lz = 1) was observed for the three isolates of *C. lusitaniae* (IMR-M-L 301, 522, and 1112) and *C. haemulonii* IMR-M-L 785.

*C. albicans* isolates produced higher lipase activity levels (mean LZ of 0.44 ± 0.06) compared to *C. haemulonii* (mean LZ of 0.73 ± 0.25) or *C. lusitaniae* (mean LZ of 0.92 ± 0.15). Differences were significant between the three species **** (*p*-values < 0.0001).

With a gradual increase in NaCl concentrations, yeast cells were inoculated in SD broth to evaluate resistance to osmotic stress. All of the tested isolates were able to grow at a concentration of 15% NaCl, including the reference strain *C. albicans* ATCC 24433. All of the strains were considered to be resistant to osmotic stress.

### 3.5. Galleria Mellonella Survival Assay

To determine in vivo infection, *G. mellonella* larvae were employed as an animal model for a five-day study period (Figure 3). No significant difference in the survival rate of *G.mellonella* larvae infected with all three Candida species was observed 4 days post infection. Survival of 50% for all three species was four days. Five days after the inoculation, larvae infected with *C. albicans* strains showed a higher mortality rate compared to those infected with *C. lusitaniae* and *C. haemulonii*, with significant differences ** (*p* < 0.0042). No larval fatalities were observed in the control groups.

## 4. Discussion

AMR is one of the top 10 global public health deteriorates. Antifungal resistance is an increasing problem involving yeast fungi like *Candida* and allied genera. In recent years, international emphasis has been placed on the emergence of *C. auris* and its multi-resistance, but less attention has been paid to other yeasts with this same feature. Currently, 20–30% of candidemia cases involve species with intrinsic or cross-resistance to scarce disposal antifungals [32,33]. This work highlights the virulence factors of *C. haemulonii* and *Clavispora lusitaniae*, since this knowledge may be associated with their emergence and provide new targets to combat their drug resistance in the near future. Likewise, *C. albicans* isolates were included to evaluate the behavior of these factors in isolates of this species with low susceptibility.

With the exception of the ATCC quality control strain, according to the CLSI M27M44S clinical breakpoints for *C. albicans*, all of the isolates presented low susceptibility to fluconazole, one of the most widely used antifungal agents. On the other hand, according to the ECV for antifungal susceptibility testing, high MIC values were obtained for all *C. haemulonii* isolates but there were variable results for *C. lusitaniae*, which may reflect the fact that this yeast population is non-wild-type.

The antifungal susceptibility profiles of members of the *C. haemulonii* complex are of concern. The low susceptibility of *C. haemulonii* to currently available antifungal drugs was also observed in our results. *C. haemulonii* complex isolates evaluated in this study showed higher MICs for fluconazole and amphotericin B than *C. lusitaniae* and *C. albicans* isolates. For fluconazole, MIC values ranging from 4 to >64 µg/mL were obtained. The fluconazole ECV established by CLSI for *C. haemulonii* is very high (128 µg/mL) and may be an indication of intrinsic resistance or of limited susceptibility to this agent. The CLSI supplement M57S, 2022, states that MIC values lower than the ECV do not imply that the isolate is susceptible to fluconazole [25]. These results agree with other authors who have reported an increase in the fluconazole resistance of this fungal complex for many years [34,35,36,37,38,39].

With MIC values higher than ECV, 3/4 studied *C. lusitaniae* isolates can be distinguished as being non-wild-type strains. Heterogeneity in fluconazole resistance largely caused by the presence of at least 12 different alleles of MRR1, which encodes a drug resistance regulator, was explained [40]. The capability of *C. lusitaniae* to rapidly develop resistance to multiple antifungals during therapy was recognized. Facilitated by the haploid nature of *C. lusitaniae*, the selection of candins and/or azole-resistant isolates following candins and azole or their combination treatments was reported, but cross-resistance to amphotericin B resistance and candins also occurred without ongoing exposure to amphotericin B [41,42,43]. The M27-A3 document states that *Candida* species with MICs >1 µg/mL are likely resistant to AMB. Our *C. lusitaniae* isolates showed an MIC range for amphotericin B from 1 to >16, confirming the low susceptibility reported for this species. *C. lusitaniae* is not intrinsically resistant to amphotericin B, but some reports describe that phenotype resistance can only be observed using agar gradient strips and cannot be detected using broth microdilution methods. This could explain the ranges of amphotericin MIC values obtained for this species [25].

Many studies have emphasized that an increase in drug resistance can be associated with several virulence factors in *Candida* [16,44,45,46,47]. In particular, studying the genes that are correlated with drug resistance and virulence factors could represent a way to resolve this problem. The literature on the behavior of these emerging drug-resistant species is scarce for *C. haemulonii* and *C. lusitaniae*.

When developing new antifungal strategies, knowledge of CSH is also essential. This biophysical parameter influences cell–cell and also cell–surface interactions, playing a crucial and complex role in the processes of virulence, as well as response to therapies. Differences in CSH may cause variable efficacy of antifungal treatments [48]. *Candida* isolates with differences in CSH may possess altered lipid metabolism and consequently may differ in their response to treatment [49]. In this study, isolates of *C. albicans* and *C. haemulonii* with low MIC values with regard to fluconazole and amphotericin B showed differences in CSH in a range between 20% and 80% and 35% and 80%, respectively. Hydrophobic *C. albicans* strains showed more resistance to fluconazole due to ergosterol overproduction and ERG11 gene overexpression, as well as overproduction and higher activity of the Cdr1 transporter [49]. While *C. albicans* and *C. haemulonii* showed higher CHS values, low values ranging between 5 and 15% were obtained for *C. lusitaniae*. Our isolates were obtained from patients who had had several previous treatments. The CSH could be reduced by antifungal therapies [48]. Further exploration of the mechanisms through which multiple drugs reduce CSH could explain additional pathways involved in pathogenesis and lead to the development of novel antifungal therapeutic strategies.

Adhesion is considered the first step in the development of biofilm formation, a significant virulence factor described for *Candida* and yeast-like fungus with medical inference. Moreover, *Candida* is adept at adhering and this process is favored, in part, by the hydrophobic interactions between yeast cells and surfaces [50,51]. Despite this low CSH values were obtained for *C. lusitaniae* in this study, high adherence performance was observed for all of the strains studied. Our results contrast those of Muadcheingka et al., 2015, who reported that it was more hydrophobic but less adherent [52]. It is well known that other factors, different to CHS, are related to adherence [13]. Furthermore, and related to this ability to adhere, all three *Candida*/*Clavispora* species tested demonstrated their capability to produce biofilm; in particular, with *C. haemulonii* and *C. albicans*, the latter is already cited as being one of the most adherent species. The ability of the *C. haemulonii* species complex to form biofilm on different types of surfaces was reported [5,38,43,53]. Although all *C. lusitaniae* isolates were adherent, they showed more interspecies variability in terms of biofilm formation. These low properties correlate with the lower virulence reported for *C. lusitaniae* than the other species [13]. In a biofilm, microorganisms are in a stable environment where they can tolerate high concentrations of antimicrobials. Moreover, biofilms’ exposure to antimicrobial drugs usually results in the induction of resistance genes. For these reasons, biofilm is a recognized virulence trait directly associated with enhanced antimicrobial resistance [54,55,56].

Lipases are enzymes of pharmacological interest because they can act as virulence factors in several infectious diseases. The role of lipases in the virulence of *Candida* has been shown to favor morphological transition, colonization, cytotoxicity, and penetration, thus enhancing their survival within the host [57]. Although most of the isolates studied in this work showed lipase activity, no uniformity in the expression of this activity for any of the three species was observed. In contrast, all of the isolates were considered to be resistant to osmotic stress. Exposure to NaCl or KCl produces rapid and effective adaptation to nutrients and stress, which benefits the virulence of *C. albicans* [58,59]. The ability to rapidly adapt to host-imposed stresses in varied environments contributes significantly to the survival of the fungus in host niches and its ability to cause infection [60].

The larvae of *Galleria mellonella* are now broadly accepted as being a model system for assessing the virulence of microbial pathogens. According to other reports, the virulence of non-*C. albicans* species in *G. mellonella* infection models is dependent on the fungal species. In our study, *C. albicans* showed higher virulence and produced lower survival rates than *C. haemulonii* and *C. lusitaniae*. *C. lusitaniae* was shown to be more virulent than the *C. haemulonii* species complex [9].

The emergence of multidrug-resistant *Candida* species, like the *C. haemulonii* species complex and *C. lusitaniae*, and the growing number of resistant isolates of the cosmopolitan *C. albicans* pose a threat and we are likely to witness the rise of new multidrug-resistant pathogenic yeast species in the near future. *Candida* and *Clavispora* may use different mechanisms for the evasion of the host immune system but also exhibit different virulence-related phenotypes to resist antifungal effects. Mutations were found in cell adhesion genes and biofilm formation, indicating that resistance is co-evolving with virulence [32].

The ability to adhere to and form biofilms associated with their hydrophobic capacity, to adapt to stress, and to infect in an in vivo model, observed in non-wild-type *Candida* and *Clavispora* isolates, shows their marked virulence features. The observed behavior is not general for an entire species; furthermore, virulence factors can vary depending on the strain. Since the factors that define virulence are related to the development of resistance of these fungi to the few antifungals available for clinical use, differences in the physiology of these cells must be considered to develop new antifungal therapies. Likewise, the patient’s history and previous treatments are also a point of analysis, since previous antifungal treatments alter the virulence factors of these yeasts, either decreasing or exacerbating them.

## Figures and Tables

**Figure 1 microorganisms-12-00212-f001:**
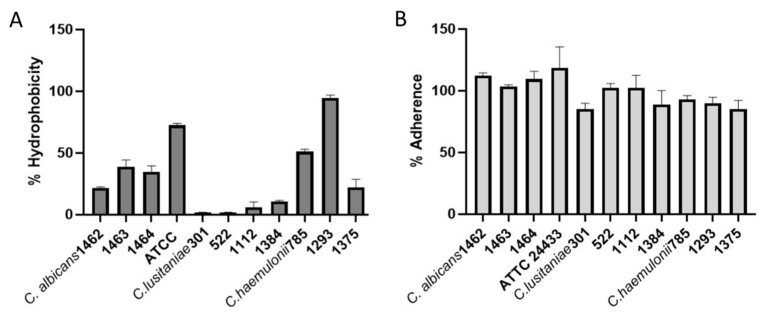
Histograms of hydrophobicity and adherence on plastic surface of *C. albicans*, *C. lusitanieae*, and *C. haemulonii* isolates. (**A**) Percentage of hydrophobicity. (**B**) Percentage of adherence on plastic surface.

**Figure 2 microorganisms-12-00212-f002:**
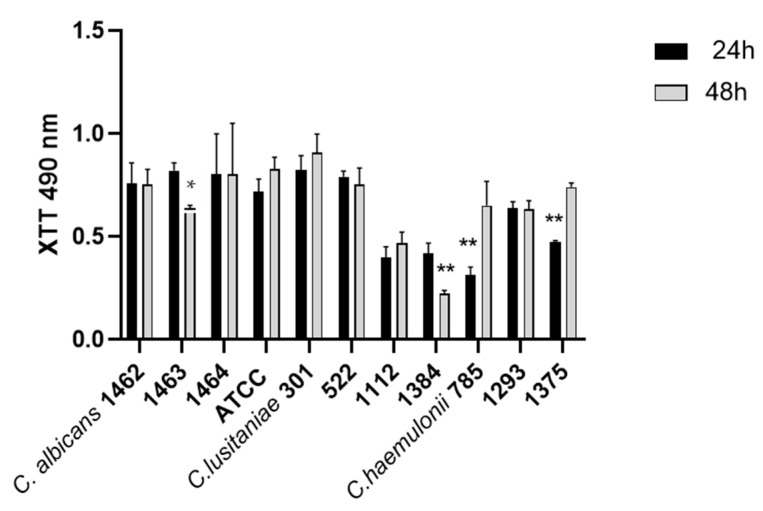
Histograms of biofilm formation after 24 and 48 h of *C. albicans*, *C. lusitaniae*, and *C. haemulonii*. * *p*-value < 0.0392, ** *p*-values < 0.004.

**Figure 3 microorganisms-12-00212-f003:**
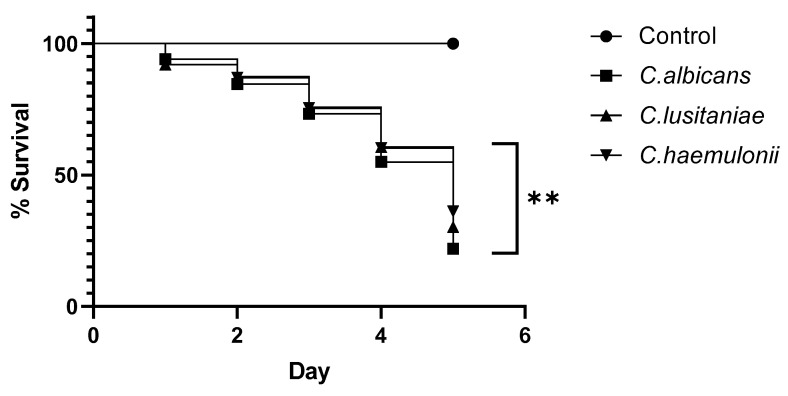
*G. mellonella* survival assays after infection with *C. albicans* (IMR-M-L 1462, 1463, 1464, ATCC 24433), *C. haemulonii* (IMR-M-L 785, 1293, 1375), and *C. lusitaniae* (IMR-M-L 301, 522, 1112, 1384). Results are expressed as % survival in comparison to uninfected (PBS-treated) larvae. Median values obtained per group (10 larvae) are presented. Larvae infected with *C. albicans* significantly decreased survival, as assessed using the Mantel–Cox log-rank test (** *p* < 0.0042).

**Table 1 microorganisms-12-00212-t001:** Antifungal activity of fluconazole and amphotericin B.

	Fluconazole	Amphotericin B
Strains	MIC (µg/mL)	MIC (µg/mL)
IMR-M-L 1462 *C. albicans*	8	0.5
IMR-M-L 1463 *C. albicans*	4	0.5
IMR-M-L 1464 *C. albicans*	4	0.5
ATCC 24433 *C. albicans*	0.25	0.5
IMR-M-L 301 *C. lusitaniae*	0.5	1
IMR-M-L 522 *C. lusitaniae*	2	2
IMR-M-L 1112 *C. lusitaniae*	2	2
IMR-M-L 1384 *C. lusitaniae*	4	>16
IMR-M-L 785 *C. haemulonii*	32	16
IMR-M-L 1293 *C. haemulonii*	>64	16
IMR-M-L 1375 *C. haemulonii*	4	2
MIC range	0.25 > 64	0.5 > 16

MIC: minimum inhibitory concentration mean.

**Table 2 microorganisms-12-00212-t002:** Lipase activity in *Candida* spp.

Strains	Lz Index	Activity	Mean LZ ± SD
IMR-M-L 1462 *C. albicans*	0.42	High	0.44 ± 0.06
IMR-M-L 1463 *C. albicans*	0.53	High	
IMR-M-L 1464 *C. albicans*	0.37	High	
ATCC 24433 *C. albicans*	0.46	High	
IMR-M-L 301 *C. lusitaniae*	1.00	None	0.92 ± 0.13 ****
IMR-M-L 522 *C. lusitaniae*	1.00	None	
IMR-M-L 1112 *C. lusitaniae*	1.00	None	
IMR-M-L 1384 *C. lusitaniae*	0.70	Moderate	
IMR-M-L 785 *C. haemulonii*	1.00	High	0.73 ± 0.25 ****
IMR-M-L 1293 *C. haemulonii*	0.70	Moderate	
IMR-M-L 1375 *C. haemulonii*	0.49	High	

Lz index: high, Lz ≤ 0.69; moderate, Lz = 0.70–0.89; weak, Lz = 0.90–0.99; none, Lz = 1. **** *p*-values< 0.0001.

## Data Availability

Data are contained within the article.

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
