# Peer review of "Identification of Virulence Factors in Isolates of Candida haemulonii, Candida albicans and Clavispora lusitaniae with Low Susceptibility and Resistance to Fluconazole and Amphotericin B"

_microorganisms, 2024, doi:10.3390/microorganisms12010212_

Round 1
Reviewer 1 Report
Comments and Suggestions for Authors
In this manuscript, the authors investigated hydrophobicity, adherence, biofilm formation, lipase activity, resistance to osmotic stress and virulence of Candida haemulonii, Candida albicans and Clavispora lusitaniae (low susceptible and re-sistant to fluconazole and amphotericin B). The results showed the virulence factors of these fungi are related to the development of resistance of antifungals available for clinical use. However, the manuscript need to be improved before considering it for publication.
1. Introduction section should be clear. Some paragraphs could be merged into one and simplified the main points, such as first two paragraphs.
2. line 104: “2 × 106 cells/ml”, “6” should superscript.
3. line 133-134: “The yeasts cells were grown as described above for 72 h at 32°C, 113
washed twice with sterile PBS and then resuspended at 37°C”, why was the incubation temperature different?
4. line 102: “yeasts were grown in SD broth at 28° C for 24 h”, line 113: “described above for 72 h at 32°C”, line 123: “incubated for 24 and 48 h at 37ºC”, line 136: “incubated at 32°C”, line 146: “incubated at 35ºC”, the authors should confirm the incubation temperature and revise.
5. line 129: “at 37°C for 2 h at 37°C”, delete “at 37°C”.
6. line 148: “2.8. Infection on Galleria mellonella model”, did the authors repeat the infection experiment? “injected with different concentration of yeast cells”, only one concentration of yeast cells was used in this study.
7. line 162, “P” should be needs to be italized.
8. line 173-192, Latin names of strains should be italicized.
9. Discussion section should be clear. Some paragraphs could be merged into one and simplified the main points,
10. line 336-341, the conclusion repeated with Abstract, the conclusion needs to be revised and more comprehensive concepts should be added there.
Comments on the Quality of English LanguageMinor editing of English language required
Author Response
- Introduction.
First two paragraphs were changed and merged.
- line 104 was corrected
- line 133-134 was correct the temperature
4.line 102 uniformed the temperature
- line 129 was corrected
- line 148 was corrected
- line 162 was corrected
- line 173-192, Latin names of strains were italicized.
- Discussion was revised
- The abstract is an extract and reflects the main article. The last paragraph was revised.
Reviewer 2 Report
Comments and Suggestions for Authors
The authors describe a great manuscript
Line 41 - Isn't Candida glabrata an emerging risk pathogen?
Line 53: Isn't this an intrinsic resistance?
Add a paragraph also talking about Candida albicans.
Line 58: Is the signal transduction pathway considered a virulence factor? Review the concept
Line 84: What about controls for other species? Please insert in results
Line 104: Review superscript values
Line 105: 500 1 octane?
Line 169: Please write all species names in italics
Line 174: This sentence is discussion. Insert in the correct section
Table 01: Insert a new column, highlighting whether the result is considered sensitive or resistant, based on the analyzed cutoff point
Line 180: How was this analysis carried out? Was it carried out between strains or between groups?
Figure 01: What would adherence above 100% be?
Figure 02: How was the data comparison carried out?
Line 250: What is the cutoff point?
Line 262: Cite robust references that support this information
No paragraph was inserted discussing the correlation between virulence findings and the sensitivity profile to the antifungals tested.
Author Response
Line 41. Candida glabrata is an emerging risk pathogen. Was included.
Line 53: Isn't this an intrinsic resistance?
Intrinsic or primary resistance is considered to be inherent (not acquired) resistance, which is a characteristic of all or almost all representatives of the species, like the intrinsic resistance of C. krusei to fluconazole. Not all representatives of C. lusitaniae possess this feature but the ability to acquire MDR after antifungal exposure and associated with the emergence of resistant or less susceptible isolates. For this reason, we do not describe C. lusitaniae as intrinsically resistant to AMB.
Line 58: Is the signal transduction pathway considered a virulence factor? Review the concept
Many of the signaling pathway interactions responsible for the physiological adjustment of the fungus to changing external conditions are species-specific and mediate the host-pathogen relationship. However, the term was removed from the list.
Line 84: What about controls for other species? Please insert in results
Candida albicans ATCC is always included as controls in all works. To obtain comparable results with other authors, we included this control.
Line 104: was corrected
Line 105: insert symbol
Line 169: Latin names of strains were italicized.
Line 174: This sentence was transferred to discussion.
Line 180: specified in the test.
Figure 01: 3 clinical isolates of C.albicans slightly exceed 100%.
Figure 02: a statistical analysis was carried out between the different species.
Line 262: Cite robust references that support this information.
Paragraph was rewrite and discussed according to CLSI states. Reference was included